# Comparison of the Chemical Components, Efficacy and Mechanisms of Action of *Chrysanthemum morifolium* Flower and Its Wild Relative *Chrysanthemum indicum* Flower against Liver-Fire Hyperactivity Syndrome of Hypertension via Integrative Analyses

**DOI:** 10.3390/ijms232213767

**Published:** 2022-11-09

**Authors:** Yue Wang, Yangyu Li, Wei Guo, Xiao Yang, Jiameng Qu, Mang Gao, Shuting Chen, Jiangru Dong, Qing Li, Tiejie Wang

**Affiliations:** School of Pharmacy, Shenyang Pharmaceutical University, 103 Wenhua Road, Shenyang 110016, China

**Keywords:** *Chrysanthemum morifolium* flower, *Chrysanthemum indicum* flower, liver-fire hyperactivity syndrome of hypertension, network pharmacology, molecular pharmacology, metabolomics, traditional Chinese medicine

## Abstract

To clarify the differences in the clinical application scope of *Chrysanthemum morifolium* flower (CMF) and *Chrysanthemum indicum* flower (CIF), two herbs of similar origin, an integrated strategy of network pharmacology, molecular pharmacology, and metabolomics was employed, with a view to investigating the commonalities and dissimilarities in chemical components, efficacy and mechanisms of action. Initial HPLC-Q-TOF-MS analysis revealed that CMF and CIF had different flavonoid constituents. The biological processes underlying the therapeutic effects of CMF and CIF on liver-fire hyperactivity syndrome of hypertension (LFHSH) were predicted to be related to inflammatory response, fatty acid production, and other pathways based on network pharmacology analysis. ELISA, molecular docking, Western blot, and metabolomics techniques showed similar effects of CMF and CIF in lowering blood pressure, resistance to tissue, organ and functional damage, and dyslipidemia. However, distinct effects were found in the regulation of inflammatory response, PI3K-Akt and NF-κB signaling pathways, lipid anabolism, renin-angiotensin system, and metabolic abnormalities. The comparable efficacies of CMF and CIF, despite having distinct mechanisms of action, may be attributed to the integration and counteraction of their different regulating capabilities on the above anti-LFHSH mechanisms. This study offers a vital platform for assessment of differential and precise applications of herbs of close origin with similar but slightly different medicinal properties, and provides a research strategy for bridging Chinese medicine and modern precision medicine.

## 1. Introduction

Essential hypertension (EH) is a cardiovascular disorder characterized by high systemic arterial pressure as the primary clinical manifestation [1], which is a pivotal risk factor for cerebral stroke, coronary atherosclerotic heart disease, and renal function impairment [2]. EH is associated with extremely high mortality and disability rates and involves complex molecular mechanisms, including activation of the sympathetic nervous system, disorder of the renin-angiotensin system (RAS), genetic tendency, environmental factors, inflammation, metabolic disorders and several other pathways that remain to be established [3,4,5]. Additionally, significant individual variations exist among EH patients. Therefore, strategies for management of EH should focus on precise screening of etiology to achieve the purpose of personalized treatment. The theory of traditional Chinese medicine (TCM) states that EH belongs to the category of vertigo. The most prevalent type is liver-fire hyperactivity syndrome of hypertension (LFHSH), which is caused by liver fire due to overwork, long-term mental stress, and improper diet [6]. The development of effective therapeutic drugs to improve the quality of life and survival rates of patients with LFHSH currently remains an unmet clinical challenge.

TCM which incorporates a wide range of targets and less resistance has considerable advantages over Western medicine [7]. *Chrysanthemum morifolium* flower (CMF) and its wild relative, *Chrysanthemum indicum* flower (CIF), represent the dried flower heads of *Chrysanthemum morifolium* Ramat. and *Chrysanthemum indicum* L., respectively, with reported functions of clearing heat, detoxification, calming the liver and improving eyesight. As medicine and food homology (MFH) materials [8,9], CMF and CIF are used as basic ingredients to produce a variety of food supplements, health foods, and treatments for redness, swelling, heat, and pain resulting from liver-fire hyperactivity syndrome (LFHS), including wind-heat syndromes of common cold, hypertension, and headache. Since the liver fire, brought on by the hyperactive Yang vital energy in the liver, will circulate throughout the body with vital energy and blood as the carrier, burning blood vessels and further triggering LFHSH, the treatment of LFHSH should be centered on calming the liver and clearing the liver fire. Thus, CMF and CIF with superior heat clearing characteristics have been recommended as optimal treatment choices [10,11,12]. Modern pharmacological experiments have consistently validated the beneficial activities of CMF and CIF in lowering blood pressure, decreasing blood lipids, and protecting against heart damage. According to earlier findings, CMF is effective in reducing blood pressure and preventing hypertension-related myocardial fibrosis [13]. Simultaneously, the CIF extract is reported to ameliorate hypertension symptoms by suppressing the inflammatory response induced by nuclear factor-κB (NF-κB) and mitogen-activated protein kinase [14]. In addition, various chemical components of CMF and CIF, such as luteolin and linarin, show activity in inhibiting the level of angiotensin II (Ang II), leading to lower blood pressure [15]. Notably, while both CMF and CIF are members of the *Compositae* family and *Chrysanthemum* genus and share many microscopic similarities, their components and therapeutic properties are distinct. Therefore, they should not be confused and substituted for each other. Nevertheless, no clear boundaries currently exist in terms of scope of application of the two herbs. In addition, blindly supplementing CMF or CIF, which are TCM that can be taken as food, can increase safety risks due to the uncertainty mechanisms. Comparative analysis of CMF and CIF in terms of chemical composition, efficacy, and mechanisms of action via modern analytical methods therefore presents important means of promoting the differential and precise application of the two herbs and standardizing the scope of their clinical application.

Elucidation of specific mechanisms within a complex biological matrix remains a significant challenge due to the diverse chemical composition of TCM Network pharmacology, commonly used for the accurate prediction of drug targets and for determination of mechanisms of TCM through the construction of multivariate networks, has recently emerged as a hot spot in the frontier of TCM research [16]. However, false positives inevitably exist in the prediction results of network models. Therefore, the utilization of multivariate analysis approaches to validate the prediction results is crucial to guarantee the reliability of data. Molecular docking analysis of interactions between drugs and target proteins through computer simulation can be effectively employed to verify the binding affinities of active compounds to key targets and enhance the reliability of network models [17]. Furthermore, metabolomics is widely used to establish physiological mechanisms through the evaluation of dynamic changes in all small-molecule metabolites in organisms before and after stimulation or disturbance by external factors [18].

In this study, *Zingiber officinale* Roscoe, *Cinnamomum cassia* (L.) J.Presl, concocted *Aconitum carmichaelii* Debeaux decoction, as well as N(gamma)-nitro-L-arginine methyl ester hydrochloride (L-NAME hydrochloride) were utilized to establish the classical LFHSH model [19]. On this basis, the in vivo and in vitro components, as well as the efficacy and mechanisms of action of CMF and CIF against LFHSH, were compared by combining network pharmacology, molecular pharmacology, and metabolomics data. HPLC-Q-TOF-MS was initially employed to determine the differences in compositions of the components between CMF and CIF in vitro and in vivo. Based on the in vivo prototype components, putative anti-LFHSH mechanisms were predicted via network pharmacology. Ultimately, in keeping with the concept of a holistic perspective, blood and tissue samples of the liver, heart, spleen, lung, and kidney were collected and ELISA, molecular docking, Western blot, and metabolomics techniques were adopted to conduct comprehensive comparative analyses of the efficacy and mechanisms of action of CMF and CIF against LFHSH, with the aim of clearly establishing the clinical application scope of the two herbs.

## 2. Results

### 2.1. Comparison of the Chemical Constituents of CMF and CIF In Vitro and In Vivo

The chemical components of CMF and CIF in vitro were qualitatively examined using a combination of multi-fragment ion filtration and neutral loss filtration technologies (Appendix A). A total of 91 compounds were identified in CMF and CIF, including 56 flavonoids, among which seven were unique to CMF and two to CIF. In addition, 28 organic acids, six amino acids, and one other compound, specifically, adenosine, were shared by the two herbs (Figure 1 and Appendix A).

Next, the peak search algorithm, isotope pattern matching, and metabolite search method, based on specific product ion or neutral loss, were utilized to screen for ions absent in the C group but present in the treatment groups to determine the bioactive constituents of CMF and CIF in plasma (Appendix A). A total of 14 prototype components were shared by the two herbs (Figure 1 and Appendix A) and 123 metabolites were identified, among which 30 and 15 were unique to CMF and CIF, respectively (Figure 1 and Appendix A).

### 2.2. Network Pharmacology Predictive Analysis

Through component target prediction platforms, a total of 77 targets of 14 prototype compounds were obtained. The component targets were intersected with 4281 disease targets from the disease target prediction databases. Intersection targets with a degree value of <4 provided by PPI analysis were deleted, finally resulting in a total of 31 targets (Appendix A). Subsequent GO enrichment and KEGG pathway analyses showed that the anti-LFHSH mechanisms of CMF and CIF were mainly associated with the regulation of PI3K activity, Akt viability, NF-κB activity, the inflammatory response, the response to interleukin-1 and other biological processes as well as cancer, lipid and atherosclerosis, TNF signaling, T-cell receptor signaling, antifolate resistance, and other pathways (Figure 2A,B).

To further screen for potential medicinal components, the “14 components−24 genes−29 pathways” multivariate network was visualized via Cytoscape (Figure 2C) and the average degree, a common parameter associated with multiple networks, was applied to determine interactions between the components. The average degree value of the 14 components was 3.29. Eight of the 14 compounds (acacetin, apigenin-7-O-glucoside, diosmetin, eriodictyol-7-O-glucuronide, hesperetin-7-O-glucuronside, luteolin-7-O-glucoside, luteolin, luteolin-7-O-glucuronside with degree values ≥ 3; Appendix A) were considered as potential components and were used for subsequent validation of the molecular mechanisms of CMF and CIF against LFHSH.

### 2.3. Blood Pressure Regulation Activity

As shown in Figure 3A,B and Appendix A, SBP, MBP, and DBP fluctuation ranges in the M group were higher than those in the C group (*p* < 0.05), indicative of successful modeling. Compared with the M group, all treatments were able to reduce blood pressure (*p* < 0.05), but no obvious discrepancies were evident between CMF, CIF, and positive drug groups (*p* > 0.05).

### 2.4. HR Measurements

As demonstrated in Figure 3C, the HR of LFHSH rats was considerably higher than that of the C group (*p* < 0.05). CMF, CIF, and positive drug treatments contributed to the attenuation of increased HR (*p* < 0.05), with no significant differences among groups (*p* > 0.05).

### 2.5. Assessment of Resistance to Functional and Organ Damage

#### 2.5.1. Efficacy in Resistance to Liver and Kidney Function Damage

The plasma levels of liver function indexes AKP, ALT, and AST (Figure 4A) and kidney function indicators BUN and CRE (Figure 4B) in the M group were higher than those in the C group (*p* < 0.05). All drug interventions led to a marked reduction in the levels of these parameters (*p* < 0.05). The results clearly indicate that LFHSH triggers liver and kidney injury, a process which is effectively counteracted by CMF, CIF, and the positive drug in a dose-dependent manner. However, the effects of the CMF and CIF were indistinguishable (*p* > 0.05).

#### 2.5.2. Efficacy in Reversing Dyslipidemia

As shown in Figure 4C, CMF, CIF, and the positive drug could restore the levels of TG, T-CHO, and LDL-C, indicating successful interference with the pathological process of dyslipidemia (*p* < 0.05). No significant differences between the effects of CMF and CIF were observed (*p* > 0.05).

#### 2.5.3. Histopathological Examination of Liver, Heart, Spleen, Lung, and Kidney Tissue

H&E staining was employed to examine pathological alterations in diverse tissues (Figure 5). Compared with the C group, liver tissue of LFHSH model rats showed obvious congestion of hepatic sinusoids, accompanied by inflammatory cell exudation and apparent cell swelling. Heart tissue displayed evident myocardial cell swelling, coupled with distinct inflammatory cell infiltration and myocardial cell arrangement disorder. The boundary between the red and white pulp of spleen tissue was unclear and the splenic corpuscles were blurred. In lung tissue, the alveolar structure was destroyed, the alveolar cavity was infiltrated by numerous inflammatory cells, and the alveolar wall was significantly thickened. In addition, the tubule epithelium of kidney tissue was markedly swollen, inflammatory cells were widespread, and a proportion of tubules displayed tubular degeneration. Following drug treatments, all histopathological characteristics were improved.

### 2.6. Influence on Inflammatory Cytokines and Key Enzymes in LFHSH Rats

To validate the network pharmacology finding that CMF and CIF exert their effects on LFHSH by alleviating the inflammatory state, the activities of inflammatory factors and key enzymes were analyzed. TNF-α (Figure 6A), IL-1β (Figure 6B), IL-4 (Figure 6C), IL-6 (Figure 6D), IL-10 (Figure 6E) and COX-2 (Figure 6F), which are crucial downstream molecules of PI3K-Akt and NF-κB signaling pathways, were abnormally secreted in the tissues and plasma of rats of the M group (*p* < 0.05). Following drug intervention, inflammatory pathology in the CMF, CIF and positive control groups was markedly relieved (*p* < 0.05). Notably, the anti-inflammatory effect of CIF was more potent than that of CMF (*p* < 0.05), and CIF showed greater ability than the positive drug to regulate COX-2 in the kidney and heart (*p* < 0.05).

### 2.7. Assessment of Proteins Associated with NF-κB and PI3K-Akt Signaling Pathways

To certify the importance of NF-κB and PI3K-Akt signaling pathways in the pathogenesis of LFHSH, both ELISA (quantitative detection) and Western blot (qualitative detection) were utilized for cross-validation with a view to reliably analyzing the expression patterns of the protein involved in the above pathways. Based on the results, NF-κB p65 and its phosphorylated form in the M group were activated concomitantly with enhanced degradation of IκBα (*p* < 0.05) (Figure 7). CMF, CIF, and the positive drug exerted anti-inflammatory effects by suppressing activation of the NF-κB signaling pathway (*p* < 0.05). Among the treatment groups, the effects of CIFH were superior to those of CMFH (*p* < 0.05). Furthermore, the PI3K-Akt signaling pathway in the M group was significantly inhibited (*p* < 0.05), while the activities of PI3K, p-PI3K, Akt, p-Akt, and FOXO1 after drug treatment were significantly restored (*p* < 0.05). Overall, CMF, CIF and the positive drug effectively upregulated the PI3K-Akt signaling pathway to ameliorate the inflammation state and lipid synthesis disorders (*p* < 0.05), but the effects of the three interventions were indistinguishable (*p* > 0.05).

### 2.8. Regulation of RAS

Molecular docking and ELISA experiments were conducted to explore the mechanisms of action of CMF and CIF against LFHSH from the perspective of RAS regulation. Losartan, valsartan, captopril and lisinopril, serving as Ang II type 1 receptor blockers (ARBs) or ACE inhibitors (ACEI) [19,20], were employed for semi-flexible docking with 6OS2 or 1O86, respectively. The docking results revealed that ALA1244, GLN1257, ASN1294, ILE1245, as well as LYS511, TYR520, GLN281, HIS353, and GLU384 residues were the primary active binding sites for 6OS2 and 1O86, respectively. Interactions between the components and receptor proteins were further assessed. All components could form hydrogen bonds with the active sites (Figure 8A,B), and the binding energies of six of the eight components—specifically, acacetin, diosmetin, hesperetin-7-O-glucuronside, luteolin, luteolin-7-O-glucoside, luteolin-7-O-glucuronside—with 1O86 and 6OS2 were less than −5 kcal/mol (Appendix A). This implies that these compounds simulate the effects of ARBs and ACEI and inhibit overactivation of RAS, which could underlie the therapeutic activities of CMF and CIF against LFHSH. In ELISA experiments, the expression of ACE (Figure 8C) and Ang II (Figure 8D) was elevated in multiple tissues and plasma of LFHSH rats (*p* < 0.05). Treatment with CMF, CIF and the positive drug effectively suppressed the levels of these indicators in a concentration-dependent manner (*p* < 0.05). Notably, the effects of CIF were significantly stronger than those of CMF (*p* < 0.05), thus supporting the superior activity of CIF in terms of regulatory effects on RAS.

### 2.9. Non-Targeted Metabolomics Analysis

To further explore the mechanisms of action of CMF and CIF against LFHSH, a non-targeted metabolomics study of plasma samples was performed via HPLC-Q-TOF-MS, with proven precision, stability, and reproducibility (Appendix A). As shown in Figure 9A,B, plasma samples of C, M, P, CMFH and CIFH groups were separated significantly in OPLS−DA score charts without overfitting behavior (Appendix A). The close distance between each administration and the C groups suggests that endogenous components in the M group are altered under pathological conditions, and that drug therapy interferes with the occurrence of these variations.

According to determinations of *p* value (*p* < 0.05), FC value (FC > 2 or FC < 0.5), and VIP value (VIP > 1), 41 biomarkers displaying significant differences between the C and M groups were screened (Appendix A) and further visualized via cluster and heat map analyses (Figure 9C). MetaboAnalyst was further used for pathway analysis of the 41 biomarkers (Figure 9D). The top 10 metabolic pathways associated with pathogenesis of LFHSH were arachidonic acid metabolism, primary bile acid biosynthesis, biosynthesis of unsaturated fatty acids, phenylalanine metabolism, arginine biosynthesis, nicotinate and nicotinamide metabolism, histidine metabolism, TCA cycle, sphingolipid metabolism, and folate biosynthesis. Alterations in endogenous components after drug intervention were comprehensively examined. A total of 19 biomarkers in the CMF, CIF and P groups were significantly restored (*p* < 0.05; Appendix A). Notably, CMF demonstrated stronger regulatory effects than CIF on five metabolites (*p* < 0.05) mainly involved in the TCA cycle, glyoxylate and dicarboxylate metabolism, histidine metabolism, pyruvate metabolism, fatty acid degradation, and purine metabolism (Figure 9E). Conversely, the regulatory activity of CIF was stronger than that of CMF for four metabolites (*p* < 0.05) that primarily participate in phenylalanine metabolism and folate biosynthesis (Figure 9F). The collective results indicate that CMF and CIF exert therapeutic effects against LFHSH via distinct biological processes.

## 3. Discussion

Over the course of a long history, two different medical systems, Chinese medicine and Western medicine, have emerged in the East and the West, respectively. Due to their origins in different cultures, Chinese and Western medicine are different in nature. In contrast to Western medicine, which mainly uses modern medical instruments to detect and treat patients at the microscopic level through specific quantitative analysis and by relying on experimental test results, Chinese medicine favors the observation of human pathogenesis on a holistic, dynamic and relational level [21]. Chinese medicine is characterized by a “Holistic concept” and a practice of “Syndrome differentiation and treatment”. It operates on the belief that the human body is an organic whole, with all internal organs being physiologically interconnected, and with local pathological changes being related to the strength and weakness of the whole body’s vital energy, blood, Yin and Yang. This therefore determines that the treatment of a local lesion must start with the entirety of the body. In addition, toxic effects are a safety issue that cannot be ignored in the clinical application of drugs, and Chinese medicine can reduce toxicity and enhance efficacy by concocting and compatibility of herbs, which is also beyond the reach of Western medicine [22]. Today, with the development of systematic scientific research and modern production techniques, more and more people around the world are using traditional therapies, including Chinese medicine, in their daily lives [23]. However, unclear chemical composition and mechanisms of action, combined with the confusing misuse of similar herbal medicines and the practice of diagnosis based on experience alone may affect the efficacy and safety of herbal medicine use and limit the international development of Chinese medicine. Therefore, using modern means to clarify the chemical composition and mechanism of action of TCM, setting a reasonable scope of clinical application, and establishing diagnostic markers are undoubtedly important means to promote the modernization and internationalization of Chinese medicine.

TCM is available in a variety of forms, including numerous herbs of similar origin, with comparable properties and functions. Abundant and diverse chemical components form the basis of their efficacy [24]. In our component identification experiments, the main constituents of CMF and CIF were flavonoids and organic acids, a result which is consistent with previously reported pharmacodynamic constituents of the two herbs [25,26]. The differences in composition between CMF and CIF were attributed to the flavonoids. Several studies have reported that common components (such as luteolin, baicalin, apigenin-7-O-glucoside and naringenin-7-O-glucoside) as well as unique components (such as naringenin, taxifolin-7-O-glucoside, apigenin and acacetin-7-O-glucoside) in CMF and CIF confer antihypertensive, antioxidant, anti-inflammatory and antiviral properties through regulating NF-κB and PI3K-Akt signaling pathways, as well as RAS and metabolic disorders [27,28,29,30,31,32]. Hence, chemical differences may be the primary factor influencing variations in the mechanisms and efficacy of the two herbs. Additionally, discrepancies in metabolite species between CMF- and CIF-treated rats reflect the different modes of interaction or transformation of distinct chemical component groups of the two herbs, confirming that the diversity of chemical constituents results in different pharmacological actions.

Elevated blood pressure and increased HR are the most common external manifestations of EH [33]. In this study, we observed no differences in the regulatory effects of CMF and CIF on blood pressure and HR, indicating comparable macro-level anti-LFHSH activities of the two herbs. Functional and organ damage, induced by chronically high loads on multiple tissues as a result of long-term sustained blood pressure increase, is another typical complication of hypertension [34]. Our results showed that both CMF and CIF participate in functional regulation to promote resistance to organic tissue damage and defend against pathological injury caused by LFHSH. To further explore the capacity of CMF and CIF in protecting tissues at the molecular level, clinical indicators frequently utilized to monitor liver, kidney function, and cardiovascular and cerebrovascular damage [35,36,37] were evaluated, which revealed comparable results. Overall, CMF and CIF exerted similar anti-LFHSH effects. Accordingly, the comparison study mainly focused on the examination of differences in the mechanisms of action between the two herbs.

In recent years, network pharmacology has been widely applied as a relatively new TCM research method owing to its conformation with the systematic and holistic thinking mode of Chinese medicine [38]. The majority of research objects in previous network pharmacology studies were compounds searched in the TCM database or absorbed into the blood after the oral administration of TCM drugs in healthy rats [39,40]. Although the latter can eliminate predictive errors (false negatives or false positives) compared with the former, the results may be inconsistent with the type of component absorbed in blood under pathological conditions. In the current study, plasma prototype compounds of CMF and CIF under the LFHSH pathological state were used for network pharmacology analysis, ensuring higher authenticity and reliability of the prediction results. Prediction analysis of network pharmacology suggested that the mechanisms of action of CMF and CIF against LFHSH primarily involve the inflammatory process and lipid anabolism. Specifically, the main biological processes enriched in network pharmacology, the PI3K-Akt and NF-κB signaling pathways, are closely related to immune inflammation and lipid anabolism, which also play pivotal roles in the development of hypertension [41,42]. In view of the prediction results, further in-depth exploration of the anti-inflammatory and lipid metabolism-modulating abilities of CMF and CIF was performed.

Chronic inflammation is considered the most critical trigger for endothelial dysfunction, which is a fundamental hallmark of LFHSH target organ impairment and pathophysiology of cardiovascular disease [4]. In our experiments, both CMF and CIF were capable of suppressing abnormal inflammatory mediator secretion, but the effect of CIF was more significant, and NF-κB signaling pathway, the most typical immune-inflammatory pathway [43,44], was inhibited by CIF to a greater extent. Our results support a superior effect of CIF in alleviating inflammation due to a greater competence in regulating NF-κB signaling pathway, which may be one of the distinct anti-LFHSH mechanisms between CIF and CMF.

The PI3K-Akt signaling pathway is a vital pathway affecting glucose and lipid metabolism, immune inflammation, cell proliferation, and blood pressure regulation. Deactivation of PI3K and Akt induces inflammatory effects and increases generation of cytokines, thereby promoting vascular endothelial cell damage [44,45]. In addition, the PI3K-Akt signaling pathway is reported to control lipid production. FOXO1 is a key target protein influencing lipid anabolism in this pathway. Akt augments lipid synthesis via phosphorylation of FOXO1, preventing FOXO1 localization and suppressing its target gene expression [46]. As shown in our study, CMF and CIF were similar in terms of ability to alter the PI3K-Akt signaling pathway, which may serve as the common mechanism of action of the two herbs against LFHSH.

RAS is a humoral regulatory system composed of peptide hormones and corresponding enzymes that function to maintain blood pressure, water and electrolyte balance, and the relative stability of the human body [19,47]. Ang II is the main effector molecule in this system; it is produced by the conversion of the Ang I, that induce systemic arterial and venous constriction, which is catalyzed by ACE [48]. The Ang II content in patients with LFHSH is markedly higher than that in patients with other TCM-type EH, such as Yin deficiency with Yang hyperactivity or Yin and Yang deficiency syndrome of hypertension [49,50]. Hence, it is an effective tool to explore the mechanism of action against LFHSH from the standpoint of regulation of RAS. In this study, the active components of CMF and CIF, screened via network pharmacology, were shown to inhibit activation of RAS by selectively hindering the process of increasing vasoconstriction activity, arising from the conversion of Ang I into Ang II and blocking binding of Ang II to AT1. This capacity was stronger in CIF, as observed from ELISA quantification, which may present another differential anti-LFHSH mechanism between CIF and CMF.

In the pathophysiology of LFHSH, metabolic disorders are also pivotal. Previous studies have shown that polyunsaturated fatty acids and their downstream derivatives, such as arachidonic acid and prostaglandins, have various beneficial functions, including lowering blood pressure, anti-inflammatory and anti-tumor properties [51]. The biomarker, γ-linolenic acid, is metabolized into a series of n-6 type polyunsaturated fatty acids and derivatives, which relax blood vessels, reduce cholesterol levels and weaken the affinity of Ang II receptors [52]. Amino acid metabolic pathways, such as histidine and phenylalanine metabolism, also play important roles in the etiology of hypertension. These pathways can attenuate hypertension-induced damage by improving metabolic patterns and lowering reactive oxygen species production [53,54]. As a vital energy-yielding metabolic pathway, the TCA cycle also affects the growth of hypertensive rats. Suppression of the TCA cycle augments the levels of reactive oxygen species and leads to increased blood pressure [55]. Dihydropteroate, a critical intermediate in folate production, was also significantly decreased in LFHSH rats. The compound is converted into tetrahydrobiopterin, an oxidative reducing auxiliary factor of endothelial-derived nitric oxide synthase that is considered one of the most important suppressors of elevated blood pressure [56]. In addition, the biomarkers sphinganine 1-phosphate of sphingolipid metabolism and adenosine of purine metabolism influence hypertension pathophysiology by boosting inflammation and angiogenesis [57,58]. The results of our metabolomic analyses indicate that severe metabolic disorders are triggered in LFHSH rats. Strong regulation of the TCA cycle, glyoxylate and dicarboxylate metabolism, histidine metabolism, pyruvate metabolism, fatty acid degradation, and purine metabolism by CMF and phenylalanine metabolism and folate biosynthesis by CIF may contribute to their therapeutic effects against LFHSH, indicating that different metabolic pathways underlie the anti-LFHSH mechanisms of CIF and CMF.

The commonalities and dissimilarities between CMF and CIF activities may be summarized as follows: (1) There are different flavonoids constituents in CMF and CIF. (2) CMF and CIF exert similar effects on lowering blood pressure and HR, resisting organic tissue damage, as well as reversing disruption of liver and kidney function, and dyslipidemia. (3) CMF exhibits a stronger ability to regulate the TCA cycle, glyoxylate and dicarboxylate metabolism, histidine metabolism, pyruvate metabolism, fatty acid degradation, and purine metabolism than CIF. Conversely, CIF is superior to CMF in regulating the inflammatory response, NF-κB signaling pathway, RAS, phenylalanine metabolism, and folate biosynthesis, while the two herbs are similarly effective in regulating PI3K-Akt signaling pathway, lipid anabolism, sphingolipid metabolism, and primary bile acid biosynthesis. Based on comprehensive analysis of the experimental results, we infer that the two herbs have similar efficacies, despite possessing distinct mechanisms of action. The differences in intervention effects on measurement indicators between CMF and CIF are shown in Table 1. An overview of the similarities and differences in the chemical components, efficacy and mechanisms of action of CMF and CIF against LFHSH is presented in Figure 10.

## 4. Material and Methods

### 4.1. Materials

CMF, CIF, *Zingiber officinale* Roscoe, *Cinnamomum cassia* (L.) J.Presl, and concocted *Aconitum carmichaelii* Debeaux were purchased from Guoda Pharmacy (Shenyang, China), and certified by Professor Dong Wang (Shenyang Pharmaceutical University, Shenyang, China). L-NAME hydrochloride (PubChem CID: 135193) was provided by Sigma-Aldrich (St. Louis, MO, USA). LC-MS-grade formic acid, HPLC-grade methanol, and acetonitrile were all offered by Fisher Scientific (San Jose, CA, USA). Xylene, anhydrous ethanol, and PBS were from the Sinopharm Chemical Reagent Co., Ltd. (Shanghai, China). Hematoxylin and eosin were obtained from Aladdin Biochemical Technology Co., Ltd. (Shanghai, China). Tianma Gouteng granules were provided by Chengdu Jiuzhitang Jinding Pharmaceutical Co., Ltd. (Sichuan, China). Ultrapure water was purchased from Wahaha Group Co., Ltd. (Hangzhou, China). Interleukin-1 beta (IL-1β, SEA563Ra), tumor necrosis factor alpha (TNF-α, SEA133Ra), interleukin-4 (IL-4, SEA077Ra), interleukin-6 (IL-6, SEA079Ra), interleukin-10 (IL-10, SEA056Ra), cyclooxygenase-2 (COX-2, SEA699Ra), angiotensin converting enzyme (ACE, SEA004Ra), and forkhead box protein O1 (FOXO1, SEA764Ra) ELISA kits were supplied by Cloud-Clone Corp. (Houston, TX, USA). Ang II (ML731165-2), phosphatidylinositol 3-kinase (PI3K, ml735116-2), protein kinase B (Akt, ml730042-2), phosphorylated PI3K (p-PI3K, ML735453-2), phosphorylated Akt (p-Akt, ML730585-2), NF-κB p65 subunit (NF-κB p65, ml731196-21), phosphorylated NF-κB p65 (NF-κB p-p65, ML735152-2), and NF-κB inhibitor alpha (IκBα, ml730433-2) ELISA kits were offered by Shanghai MLBio Biotechnology Co., Ltd. (Shanghai, China). Alkaline phosphatase (AKP, A059-2-2), alanine aminotransferase (ALT, C009-2-1), aspartate aminotransferase (AST, C010-2-1), urea nitrogen (BUN, C013-2-1), creatinine (CRE, C011-2-1), total cholesterol (T-CHO, A111-1-1), triglyceride (TG, A110-1-1), high-density lipoprotein cholesterol (HDL-C, A112-1-1), and low-density lipoprotein cholesterol (LDL-C, A113-1-1) testing kits were from the Nanjing Jiancheng Bioengineering Institute (Jiangsu, China). Heptadecanoic acid (PubChem CID: 10465) and 2-Amino-3-(2-chlorophenyl)propanoic acid (PubChem CID: 85679) with a purity greater than 98% were provided by Chengdu Chroma Biotechnology Co., Ltd. (Sichuan, China).

### 4.2. Animals and Groupings

Male Wistar rats raised in the SPF Animal Experimental Center were used (Shenyang, China; quality certificate number: SCXK2020-0001). All animal procedures were authorized by the Animal Ethical Committee of Changsheng Biotechnology (IACUC No. CSE202106002). After 1 week of adaptive feeding, rats were randomly divided into control (C), model (M), low-dose (1.35 g/kg/d; equivalent dose for rats after conversion of daily dose for human use in Chinese Pharmacopoeia, 2020 edition) of CMF (CMFL) and CIF (CIFL), medium-dose (2.7 g/kg/d; two times as the human equivalent dose) of CMF (CMFM) and CIF (CIFM), high-dose (5.4 g/kg/d; four times as the human equivalent dose) of CMF (CMFH) and CIF (CIFH), and positive control Tianma Gouteng granule (1.35 g/kg/d; human equivalent dose; P) treatment groups. Overall, 16 rats were allocated to the CMFM and CIFM groups and eight rats were designated to other groups, of which eight rats in each group were selected for pharmacodynamics and mechanistic research experiments, and eight rats in the CMFM and CIFM treatment groups for plasma component analysis of CMF and CIF under the LFHSH pathological state.

### 4.3. Animal Experimental Design and Sample Collection

Modeling was performed according to an earlier protocol, with slight modifications [6]. Modeling reagents and CMF and CIF extracts were prepared by refluxing them with water as a solvent (Appendix A). Every morning, rats in the C group had a water dose of 1 mL/100 g (reagent volume/rat body weight) intragastrically administered, and rats in the other groups were treated with modeling reagents. Every afternoon, rats in the C and M groups were intragastrically administered 1 mL/100 g (reagent volume/rat body weight) water, and rats in the other groups were treated with the corresponding doses of therapeutic drugs. Both modeling and drug interventions continued for 28 d.

We collected blood from the orbit at 0.5, 1, 2, 4, 6, and 8 h from rats in the CMFM and CIFM (*n* = 8) groups after the last administration. Anticoagulant treatment was conducted to prepare plasma for component identification experiments. For pharmacodynamics and mechanistic analyses, blood samples were collected from the orbit 12 h after the final administration in all groups (*n* = 8). A proportion of samples was incubated at room temperature without anticoagulation treatment to prepare serum, and the remainder treated with anticoagulants to prepare plasma. Subsequently, all animals were sacrificed via decapitation and liver, heart, spleen, lung, and kidney tissues were immediately removed.

### 4.4. Component Identification Experiment

CMF and CIF sample solutions were prepared according to the procedures described in the Appendix A and diluted to 0.015 g/mL for in vitro component identification. For in vivo samples, plasma from eight rats in the CMFM and CIFM groups and obtained at each time-point was pooled separately. Following the addition of 3 mL methanol to 1 mL of mixed plasma for protein precipitation, the sample was vortexed for 3 min and centrifuged at 10,142× *g* for 5 min. The supernatant was dried under a nitrogen flow. The residue was subsequently dissolved in 100 μL methanol and vortexed for mixing again for 3 min, followed by ultrasonication for 5 min and centrifugation at 10,142× *g* for 5 min. The prepared samples were analyzed via HPLC-Q-TOF-MS under the specific parameters specified in the Appendix A.

The in vitro and in vivo chemical composition databases of CMF and CIF were established based on various data processing modules in PeakView 1.2.1 (PeakView software, SCIEX, Framingham, MA, USA) and MetabolitePilot 1.5 (MetabolitePilot software, SCIEX, Framingham, MA, USA), which facilitated rapid and straightforward analysis of chemical components and metabolites.

### 4.5. Network Pharmacology Analysis

Plasma prototype compounds from the component identification experiment were selected for the network pharmacology study. The component targets were predicted using the Swiss Target Prediction (http://www.swisstargetprediction.ch/ (accessed on 24 January 2022)) and PharmMapper (http://www.lilab-ecust.cn/pharmmapper/ (accessed on 24 January 2022)) platforms. TTD (http://db.idrblab.net/ttd/ (accessed on 25 January 2022)), DrugBank (https://go.drugbank.com/ (accessed on 25 January 2022)) and GeneCards (https://www.genecards.org/ (accessed on 25 January 2022)) databases were used to search for the disease targets of “Hypertension” and “Essential hypertension”. The intersecting component targets and disease targets were obtained and the STRING (https://cn.string-db.org/ (accessed on 26 January 2022)) database was utilized to construct a protein–protein interaction (PPI) network. Next, GO enrichment and KEGG pathway analyses were performed on key targets using the DAVID (https://david.ncifcrf.gov/ (accessed on 28 January 2022)) database, and the component−target−pathway network was visualized with Cytoscape 3.9.0 (Cytoscape software, UC, San Diego, La Jolla, CA, USA).

### 4.6. Hematoxylin-Eosin (H&E) Staining

Tissues of liver, heart, spleen, lung, and kidney were placed in an embedding frame, rinsed with flowing water, drained, and embedded in paraffin into wax blocks, which were baked for 4 h to prepare tissue paraffin sections. Paraffin sections were baked for a further 30 min and dewaxed in water using xylene, as well as various concentrations of ethanol and PBS solution. Ultimately, sections were sequentially stained with hematoxylin and eosin solution, dehydrated, made transparent, mounted, and photographed under a microscope (BC46 device, Olympus, Tokyo, Japan) at a 200-fold magnification.

### 4.7. Blood Pressure and Heart Rate (HR) Measurements

During the 28-day model cycle, mean arterial pressure (MBP), diastolic pressure (DBP), systolic pressure (SBP), and HR of awake rats were measured using an intelligent noninvasive manometer (BP-2010A device, Softron, Beijing, China) on days 0, 9, 18 and 28, respectively. Each index was measured three times and the average value was taken as the final result.

### 4.8. ELISA

Preparation of serum, plasma, and tissues samples of liver, heart, spleen, lung and kidney as well as viability measurements of liver function indexes (AKP, ALP, AST), kidney function indicators (BUN, CRE), blood lipid indexes (T-CHO, TG, HDL-C, LDL-C), inflammatory mediators (IL-1β, TNF-α, IL-4, IL-6, IL-10, COX-2), effector molecules of RAS (ACE, Ang II) and key target proteins (PI3K, Akt, p-PI3K, p-Akt, FOXO1, IκBα, NF-κB p65, NF-κB p-p65) were conducted using ELISA and the appropriate specific test kits.

### 4.9. Western Blot Analysis of Serum Key Target Proteins

Total serum protein was quantified using a Pierce™ BCA protein quantification kit (Thermo, San Jose, CA, USA), and all samples diluted to 2 μg prot/µL. Samples were denatured in a 100 °C metal bath for 15 min with a protein loading buffer (Epizyme, Shanghai, China) for further analysis. Sodium dodecyl sulfate–polyacrylamide gel electrophoresis from 80 V to 120 V and electric transfer at 250 mA were performed after the addition of 2.5 μL marker (Fisher, San Jose, CA, USA) and 20 μg serum samples. Subsequently, membranes were blocked with a rapid blocking solution (Genefist, Oxfordshire, UK) for 10 min, eluted with TBST (Solarbio, Beijing, China), and incubated with primary antibodies specific for p-PI3K (1:1000, ab182651; Abcam), Akt (1:1000, 4691T; CST), p-Akt (1:1000, ab278565; Abcam), NF-κB p65 (1:1000, 8242T; CST), NF-κB p-p65 (1:1000, 3033S; CST) and transferrin (serum loading control, 1:10,000, ab82411; Abcam) overnight at 4 °C. Following re-elution with TBST, bands were incubated with a secondary antibody (1:5000, bs-40295G-HRP; Bioss) for 2 h. Finally, the strips were evenly coated with ECL developer solution (US Everbright, Suzhou, China) and visualized using the Tanon 5200 Multi fully automated chemiluminescence image analysis system (Tanon, Shanghai, China).

### 4.10. Molecular Docking

Mol2 structures of bioactive components and X-ray 3D structures of Ang II type 1 receptor (PDB:6OS2) and ACE crystals (PDB:1O86) were downloaded from TCMSP (https://old.tcmsp-e.com/tcmsp.php (accessed on 1 March 2022)) and RCSB PDB protein crystal structure (https://www.pdbus.org/ (accessed on 1 March 2022)) databases, respectively. AutoDock 4.2.6 (AutoDock software, Scripps Research, La Jolla, CA, USA) was adopted to confirm the charge, protonation state, dehydration, and hydrogenation of the target protein, and the low-energy conformation of the ligand structure was subsequently satisfied. Followed by semi-flexible docking of two receptors with active constituents, the molecular docking results were visualized with PyMOL 2.4 (PyMOL software, Schrödinger, New York, NY, USA).

### 4.11. Plasma Non-Targeted Metabolomics Study

An internal standard solution containing 10 μg/mL heptadecanoic acid and 5 μg/mL 2-amino-3-(2-chlorophenyl)propanoic acid) (10 μL) and methanol (300 μL) was added to 100 μL plasma, vortexed for 3 min and centrifuged at 10,142× *g* for 5 min. The supernatant was dried under a nitrogen flow. Next, the residue was dissolved in 50 μL methanol and vortex-mixed again for 3 min, followed by ultrasonication for 5 min and centrifugation at 10,142× *g* for 5 min. Subsequently, the prepared samples were analyzed under set HPLC-Q-TOF-MS conditions (Appendix A).

Peak identification, peak matching, and internal standard normalization were processed through the XCMS (https://xcmsonline.scripps.edu/ (accessed on 6 January 2022)) platform. The MarkerView 1.2.1 (MarkerView software, SCIEX, Framingham, MA, USA) and SIMCA-P 14.1 (SIMCA-P program, Umetrics, Malmö, Sweden) were employed for statistical analysis of multivariate data. Furthermore, metabolites were identified with the HMDB (https://hmdb.ca/ (accessed on 7 January 2022)) and MetDNA (http://metdna.zhulab.cn/ (accessed on 8 January 2022)) web servers and KEGG pathway analysis was performed via the MetaboAnalyst (https://www.metaboanalyst.ca/ (accessed on 12 January 2022)) platform.

### 4.12. Statistical Analysis

All calculated experimental values are presented as mean ± SD. The Student’s *t* test was conducted using SPSS 26.0 (SPSS software, SourceForge, San Diego, CA, USA).

## 5. Conclusions

An integration strategy of network pharmacology, molecular pharmacology, and metabolomics was developed in this study to distinguish the clinical application scopes of two herbs of similar origin, CMF and CIF, based on a comparison of the chemical components, as well as the efficacy and activity mechanisms. The results clearly demonstrate comparable anti-LFHSH efficacy of CMF and CIF on a macro-level. However, these similar effects were achieved by the integration and counteraction of the different regulatory capabilities of CMF and CIF on disparate anti-LFHSH mechanisms based on different chemical components. The significant differences of indicators in the restorative effects between CMF and CIF include the key protein NF-κB p-p65 and its downstream inflammatory mediators initially identified in the inflammatory and pathway analysis, ACE and Ang II subsequently discovered in the RAS regulation study and differential metabolites, such as (S)-malate, dihydropteroate, L-palmitoylcarnitine, finally determined in the metabolomics analysis, which could be used to achieve precise therapeutic application of the two herbs in the clinic. In summary, we performed a comparative analysis of the chemical components, efficacy, and mechanisms of action of two herbs of similar origin, with analogous but slightly different medicinal properties in treatment of the same disease via modern analytical methods, which is helpful for clarification of their differential and precise scope of applications. The study described also provides a research strategy for bridging the Chinese medicine and modern precision medicine.

## Figures and Tables

**Figure 1 ijms-23-13767-f001:**
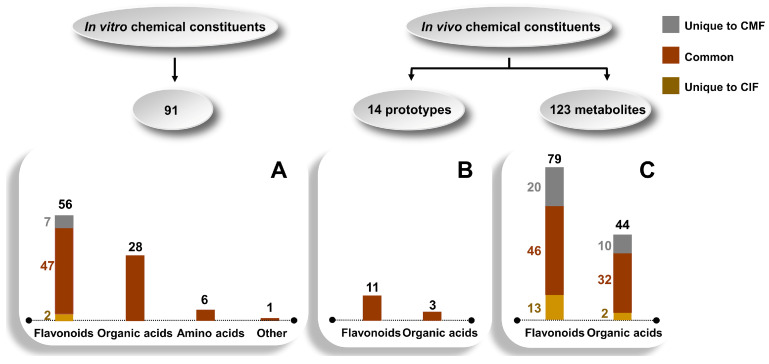
The composition of the in vitro and in vivo components between CMF and CIF. (**A**)—In vitro chemical composition differences between CMF and CIF extracts; (**B**,**C**)—Differences in prototype components and metabolites in rat plasma between CMF and CIF under the LFHSH pathological state.

**Figure 2 ijms-23-13767-f002:**
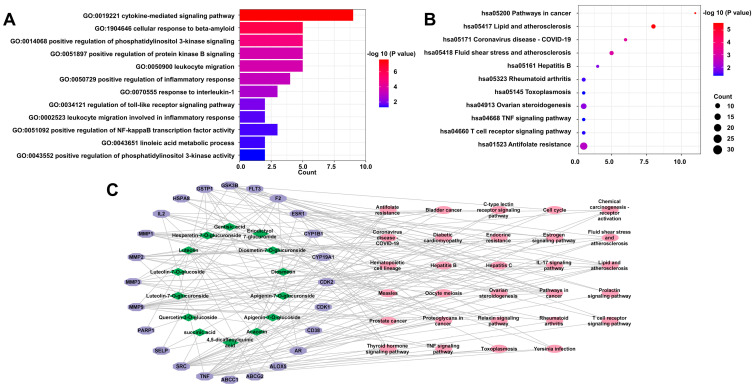
Prediction results of network pharmacology study. (**A**)—GO enrichment analysis; (**B**)—KEGG pathway analysis; (**C**)—“Component−target−pathway” multivariate network.

**Figure 3 ijms-23-13767-f003:**
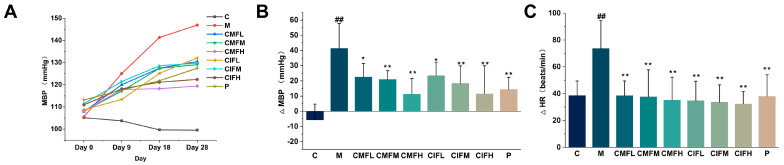
The competence of CMF and CIF in lowering blood pressure and HR. (**A**)—The change of MBP under four measurements; (**B**)—Difference value between the last and the first measurement of MBP; (**C**)—Difference value between the last and the first measurement of HR. Values shown are mean ± SD. ^##^
*p* < 0.01 compared with C group; * *p* < 0.05, ** *p* < 0.01 compared with M group.

**Figure 4 ijms-23-13767-f004:**
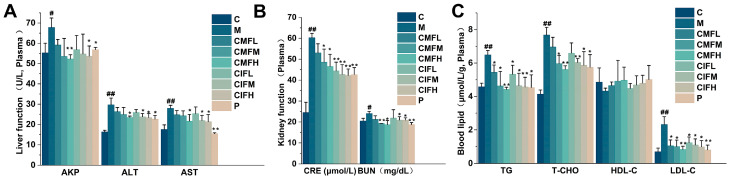
The power of CMF and CIF to resist functional damage. (**A**–**C**)—Determined results of liver, kidney function indicators, and blood lipid indexes. Values shown are mean ± SD. ^#^
*p* < 0.05, ^##^
*p* < 0.01 compared with C group; * *p* < 0.05, ** *p* < 0.01 compared with M group.

**Figure 5 ijms-23-13767-f005:**
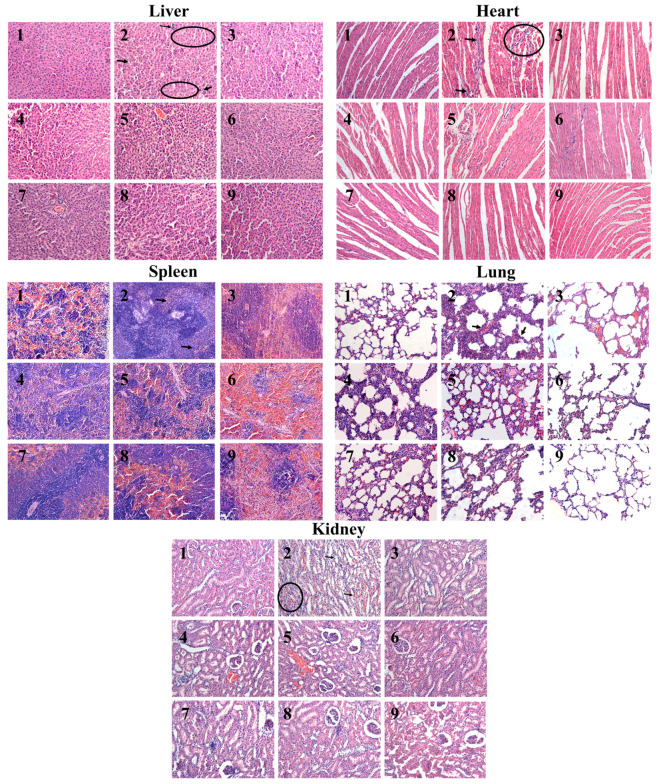
H&E staining results of tissues (200× magnification). **1**–**9** represent the C, M, P, CMFL, CMFM, CMFH, CIFL, CIFM, and CIFH groups, respectively.

**Figure 6 ijms-23-13767-f006:**
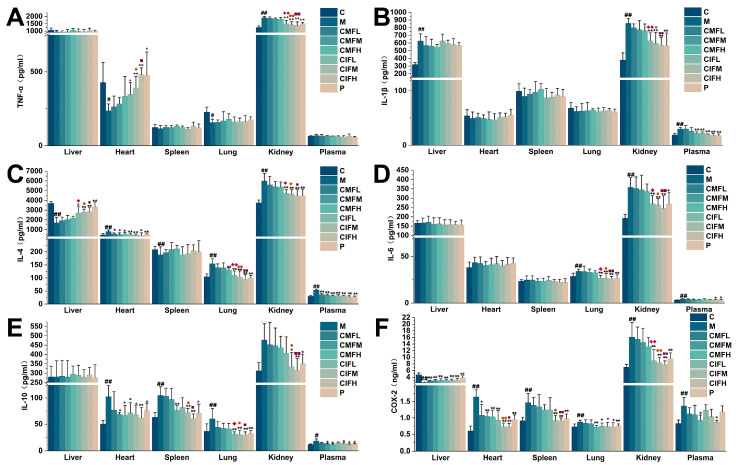
Activity assay of inflammatory mediators. (**A**–**F**)—Determined results of TNF-α, IL-1β, IL-4, IL-6, IL-10 and COX-2 in plasma, and tissues samples of liver, heart, spleen, lung and kidney. Values shown are means ± SD. ^#^
*p* < 0.05, ^##^
*p* < 0.01 compared with C group; * *p* < 0.05, ** *p* < 0.01 compared with M group; ^◆^
*p* < 0.05, ^◆◆^
*p* < 0.01 compared with CMFL group; ^●^
*p* < 0.05, ^●●^
*p* < 0.01 compared with CMFM group; ^■^
*p* < 0.05, ^■■^
*p* < 0.01 compared with CMFH group.

**Figure 7 ijms-23-13767-f007:**
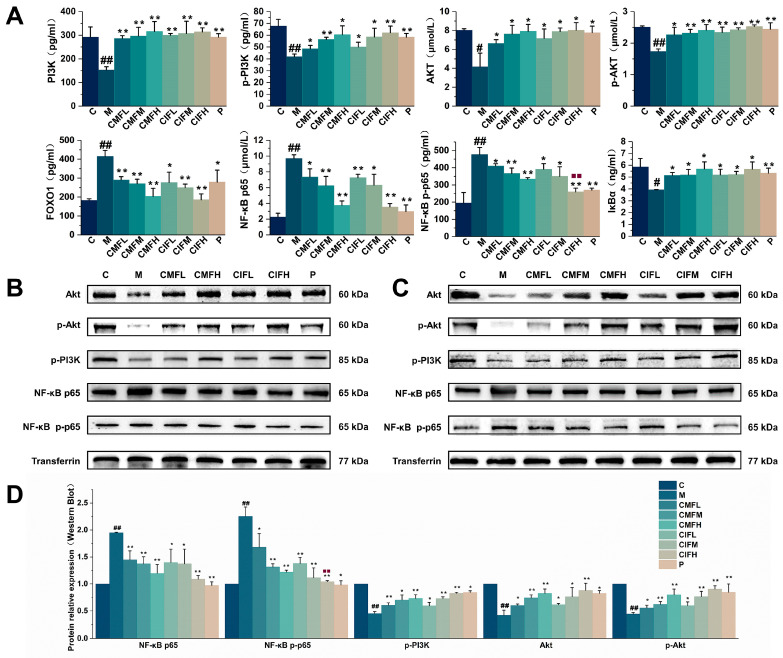
Analysis of the key proteins of NF-κB and PI3K-Akt signaling pathways. (**A**)—ELISA analysis of PI3K, p-PI3K, Akt, p-Akt, FOXO1, NF-κB p65, NF-κB p-p65, IκBα, respectively; (**B**)—Typical bands for Western blot analysis of Akt, p-Akt, p-PI3K, NF-κB p65, and NF-κB p-p65 in C, M, CMFL, CMFH, CIFL, CIFH, P groups; (**C**)—Typical bands for Western blot analysis of Akt, p-Akt, p-PI3K, NF-κB p65, and NF-κB p-p65 in C, M, CMFL, CMFM, CMFH, CIFL, CIFM, CIFH groups; (**D**)—The relative expression of proteins in Western blot analysis. Values shown are means ± SD. ^#^
*p* < 0.05, ^##^
*p* < 0.01 compared with C group; * *p* < 0.05, ** *p* < 0.01 compared with M group; ^■■^ *p* < 0.01 compared with CMFH group.

**Figure 8 ijms-23-13767-f008:**
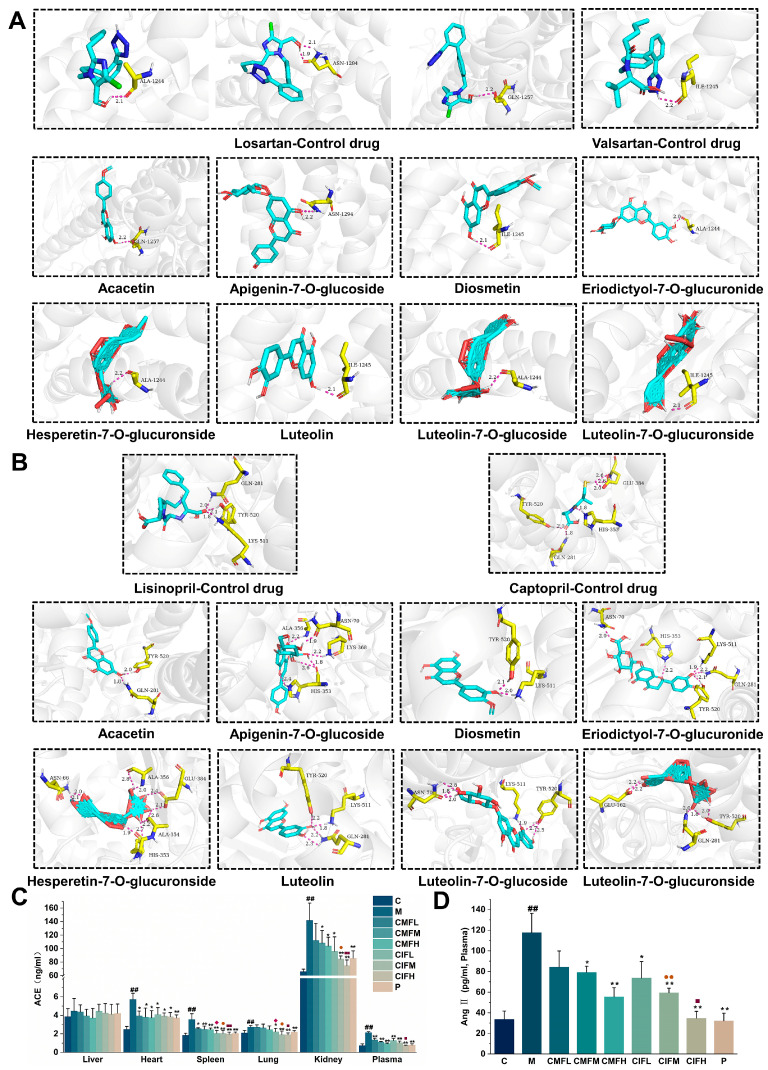
The abilities of CMF and CIF in regulating RAS. (**A**,**B**)—Molecular docking interaction diagrams of active compounds with 6OS2 and 1O86, respectively; (**C**)—ACE activity in tissues and plasma; (**D**)—Ang II viability in plasma. Values shown are means ± SD. ^##^
*p* < 0.01 compared with C group; * *p* < 0.05, ** *p* < 0.01 compared with M group; ^◆^ *p* < 0.05compared with CMFL group; ^●^ *p* < 0.05, ^●●^ *p* < 0.01 compared with CMFM group; ^■^ *p* < 0.05, ^■■^ *p* < 0.01 compared with CMFH group.

**Figure 9 ijms-23-13767-f009:**
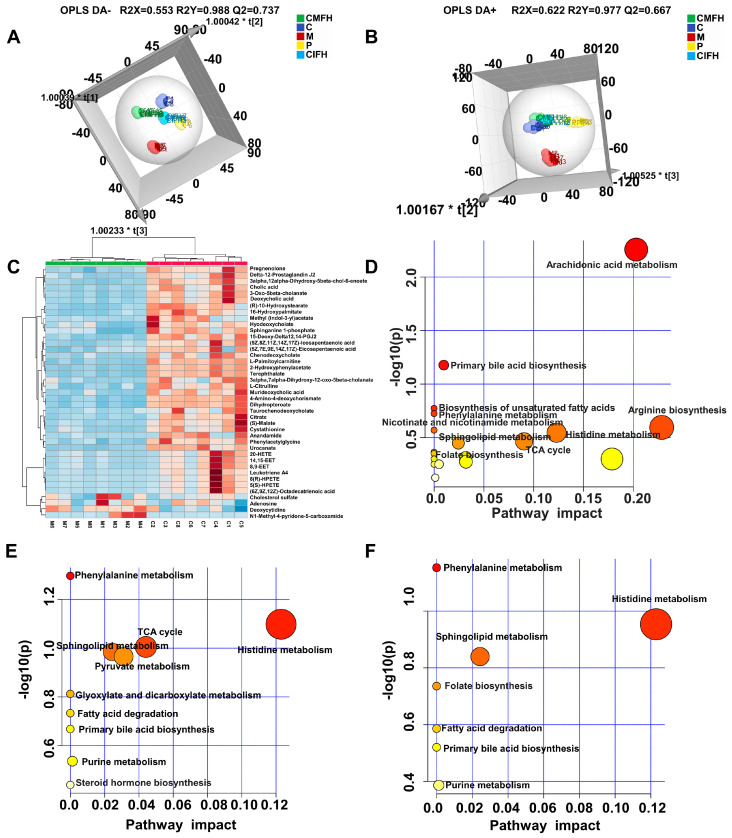
Non-targeted metabolomics investigation of rat plasma samples. (**A**,**B**)—OPLS−DA score charts under negative and positive scan modes of C, M, P, CMFH, and CIFH groups; (**C**)—Heat map of differential biomarkers between the C and M groups; (**D**)—Pathway analysis of differential biomarkers between the C and M groups; (**E**,**F**)—Pathway analysis of biomarkers called back in CMF and CIF groups, respectively.

**Figure 10 ijms-23-13767-f010:**
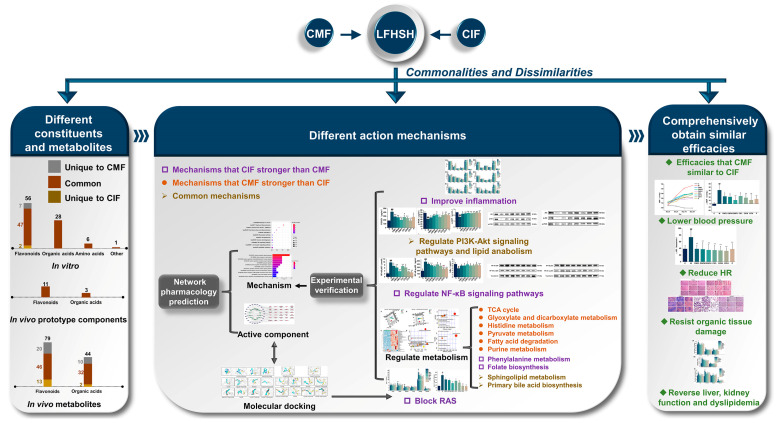
Summary of the commonalities and dissimilarities between CMF and CIF in working agaisnt LFHSH.

**Table 1 ijms-23-13767-t001:** The differences in intervention effects on measurement indicators between CMF and CIF.

Indicator	Sample	Differences in Intervention Effect		
		CMF Superior to CIF (*p* < 0.05)	CIF Superior to CMF (*p* < 0.05)	CMF Similar Effective to CIF (*p >* 0.05)
Liver and kidney function, and blood lipid indexes	Plasma	None	None	AKP, ALT, AST, BUN, CRE, TG, T-CHO, LDL-C
Inflammatory mediators	Liver	None	IL-4	COX-2
	Heart	None	TNF-α, COX-2	IL-4, IL-10
	Spleen	None	IL-10, COX-2	None
	Lung	None	IL-4, IL-6, IL-10	COX-2
	Kidney	None	TNF-α, IL-1β, IL-4, IL-6, IL-10, COX-2	None
	Plasma	None	None	IL-1β, IL-4, IL-6, IL-10, COX-2
Key proteins in NF-κB and PI3K-Akt signaling pathway	Serum	None	NF-κB p-p65	PI3K, p-PI3K, Akt, p-Akt, FOXO1, NF-κB p65, IκBα
Effector molecules of RAS	Liver	None	None	None
	Heart	None	None	ACE
	Spleen	None	ACE	None
	Lung	None	ACE	None
	Kidney	None	ACE	None
	Plasma	None	ACE, Ang II	None
Metabolites	Plasma	(S)-Malate, 15-Deoxy-Delta12,14-PGJ2, Adenosine, L-palmitoylcarnitine, Urocanate	2-Hydroxyphenylacetate, 4-Amino-4-deoxychorismate, Dihydropteroate, Terephthalate	Cholesterol sulfate, Methyl (indol-3-yl)acetate, Anandamide, Chenodeoxycholate, Hyodeoxycholate, Murideoxycholic acid, Sphinganine 1-phosphate

## Data Availability

All the data used to support the findings of this study are available from the corresponding author upon reasonable request.

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
