# Peer review of "Comparison of the Chemical Components, Efficacy and Mechanisms of Action of Chrysanthemum morifolium Flower and Its Wild Relative Chrysanthemum indicum Flower against Liver-Fire Hyperactivity Syndrome of Hypertension via Integrative Analyses"

_ijms, 2022, doi:10.3390/ijms232213767_

Round 1

Reviewer 1 Report

The authors performed a comparative analysis of the chemical components, efficacy, and mechanisms of action of two herbs of similar origin with analogous.  This paper was covered all pathway and the mechanism. This paper was excellent and will be useful. 

Author Response

Please see the attachment, Thank you!

Reviewer 2 Report

1. The authors should provide the reason about the concentration of CMF and CIF in the manuscript.

2. The authors should provide the IACUC No. in the animal test.

3. The authors should provide the detail information and pathological exchange about figure 1 in figure legend.

4.The authors should provide the catalog No. about Ang II, phosphatidylinositol 3-kinase (PI3K), protein kinase B (Akt), phosphorylated PI3K (p-PI3K), phosphorylated Akt (p-Akt), NF-κB p65 subunit (NF-κB p65), phosphorylated NF-κB p65 (NF-κB p-p65), NF-κB inhibitor alpha (IκBα) in the manuscript.

5. The authors should recheck figure legend of Figure 6. And, the authors should provide the detail procedure of Figure 6 in the manuscript.

6. The authors should add the figure legend and Y-axis of figure 7. And, the authors should remove the replay the figure of western blot.  

Author Response

Please see the attachment, Thank you!

Reviewer 3 Report

Of course, the presented study is very interesting and promising in the project of combining traditional Chinese medicine and modern precision medicine. Each of the mentioned medicines, namely the Chinese one, has long roots, its own traditions and achievements, and results. Considering that at present there is an active entry of Traditional Chinese Medicine into world medicine, it is very important to conduct evidence-based studies showing the effectiveness of treatment with herbs, minerals and animal products on a par with chemically synthesized preparations of world medicine.

In the submitted manuscript of an article in a scientific journal, it is important to pay more attention to the description of the use in traditional Chinese medicine and not only of the indicated and used plant species. Provide more evidence why the selected species were studied.

And in the discussion, give a paragraph or two - about the comparison of world practice and Traditional Chinese medicine.

In general, I would like to wish success and new excellent results, including on the way of merging Traditional Chinese medicine with the world experience in the use of medicinal plants for the treatment of various human diseases.

Author Response

Please see the attachment, Thank you!

Round 2

Reviewer 2 Report

The manuscript is good enough for publication.